# Comparison and Analysis of Detection Methods for Typhoon-Storm Surges Based on Tide-Gauge Data—Taking Coasts of China as Examples

**DOI:** 10.3390/ijerph20043253

**Published:** 2023-02-13

**Authors:** Peipei Ma, Guosheng Li

**Affiliations:** 1Institute of Geographic Sciences and Natural Resources Research, Chinese Academy of Sciences, Beijing 100101, China; 2University of Chinese Academy of Sciences, Beijing 100049, China

**Keywords:** storm surge, tide gauges, Pauta criterion, Chauvenet criterion, Pareto distribution, kurtosis coefficient, residual water level, target detection evaluation indicators, coasts of China

## Abstract

Global warming is predicted to lead to a new geographic and spatial distribution of storm-surge events and an increase in their activity intensity. Therefore, it is necessary to detect storm-surge events in order to reveal temporal and spatial variations in their activity intensity. This study attempted to detect storm-surge events from the perspective of detecting outliers. Four common outlier-detection methods, the Pauta criterion (PC), Chauvenet criterion (CC), Pareto distribution (PD) and kurtosis coefficient (KC), were used to detect the storm-surge events from the hourly residual water level data of 14 tide gauges along the coasts of China. This paper evaluates the comprehensive ability of the four methods to detect storm-surge events by combining historical typhoon-storm-surge events and deep-learning target-detection-evaluation indicators. The results indicate that (1) all of the four methods are feasible for detecting storm surge events; (2) the PC has the highest comprehensive detection ability for storm-surge events (F1 = 0.66), making it the most suitable for typhoon-storm-surge detection in coastal areas of China; the CC has the highest detection accuracy for typhoon-storm-surge events (precision = 0.89), although the recall of the CC is the lowest (recall = 0.42), as only severe storm surges were detected. This paper therefore evaluates four storm-surge-detection methods in coastal areas of China and provides a basis for the evaluation of storm-surge-detection methods and detection algorithms.

## 1. Introduction

Storm surges (residual water levels after removing long-term mean sea-level and tidal components) are natural phenomena consisting of the abnormal rise and fall in sea levels caused by strong wind and sudden changes in air pressure, accompanied by strong atmospheric disturbances, such as tropical cyclones, extratropical cyclones or other weather systems [1]. In this paper, the storm surge is equivalent to the residual water level. The residual water level, also known as abnormal water levels, is the sea-level disturbance caused by weather and other random factors, mainly including short-term water-level abnormalities caused by wind, air pressure and other weather factors, as well as seasonal sea-level abnormalities caused by climatic factors (weak tide signals generated by non-linear shallow-water effects). The residual water level can be obtained by subtracting the astronomical tide level from the observed tide level [2,3]. Compared with the residual-water level (storm surge), data errors and noise are negligible. Storm surges are generally divided into typhoon storm surges, caused by typhoons and extratropical storm surges, caused by extratropical cyclones. In China, extratropical storm surges mostly occur in spring and autumn in the northern coastal areas, with occasional occurrences in summer. In these cases, the tide-surge process is relatively gentle and the heights of these storm surges are lower than those of typhoon-storm surges. Along the southeast coast of China, however, storm surges are predominantly of the typhoon type and mostly occur in summer and autumn. Typhoon-storm surges can cause seawall collapse, house collapse, human and livestock casualties, the flooding of farmland, the salinization of land and the degradation of cultivated land within a short time, thus causing huge losses in human lives and property and in industrial and agricultural production [4]. The synchronous occurrence of storm surges and astronomical tides often causes a sharp rise in seawater levels. When a storm surge exceeds the water-warning level, a rapid surge within a short time may cause a storm-surge disaster. This is the main cause of huge disasters and losses of human life and property during storm surges.

China is one of the few countries that suffers from both tropical storm surges and extratropical storm surges. China is located on the west coast of the Pacific Ocean, with its coastal regions spanning a wide range of latitudes. In recent years, the losses caused by storm-surge disasters have ranked first among various marine disasters in China, and are increasing year by year. In the 21st century, as a result of global warming and sea-level rises, the coasts of China will face a severe threat from storm-surge disasters [5]. In this context, many new problems and challenges have arisen in the monitoring of storm-surge events. In order to protect the lives and property of citizens along the coasts of China, as well as to reduce economic losses in these areas, the most effective method is to improve the detection and forecast precision of storm surges.

Tide-gauge data are considered to be the most reliable data sources for observing storm-surge events and can be used to monitor the real tide level during a storm surge; they have been widely used for studies on global and regional storm -surge events. Scholars at home and abroad have also conducted studies on the detection of storm-surge events by using tide-gauge data; however, the detection thresholds and criteria are not uniform. For example, Zhang et al. [6] chose surges greater than two standard deviations of the annual residual-water-level data as a criterion to determine a major storm event on the east coast of the United States. Zhang et al. [6] reported that although the choice to use two standard deviations is somewhat arbitrary, it served to focus their analysis on the largest, most damaging storms and achieved good results. Ji et al. [7] and Taqi et al. [8] used three standard deviations as the threshold criteria for detecting storm surges, without explaining the reason for their selection. Dolan and Davis [9] used a maximum significant wave height of 1.5 m as a criterion to delineate storms, but did not present clear physical reasons for why this value was chosen. Additionally, there is another problem with using wave data. Distant storms can produce large swell waves unaccompanied by damaging winds or storm surges, resulting in an exaggeration of the number of coastal storms. MacClenahan et al. [10] developed an algorithm to recognize a storm surge by applying a set of wind thresholds over a defined period of time. The algorithm was designed to determine the start time and total duration of storm surges. Atkinson et al. [11] also developed an algorithm to recognize storm surges by using strong wind events. By using the direct correlation between strong-wind-observation data and a strong-storm system, they further assumed that the strong-wind-observation data were also directly correlated with the storm surge events; they then identified the storm events from the time-series wind observations. Adam et al. [12] improved the strong-wind-recognition algorithm developed by Atkinson et al. [11] to identify storm-surge events from continuous time-series water-level data. According to the research of the above scholars, the key to recognizing and detecting storm-surge events is to acquire a threshold criterion from continuous time-series wind data or water level data and then detect the storm-surge events. However, the threshold criteria necessary for the identification of storm-surge events are not uniform at present. Furthermore, the threshold criteria for the points mentioned above are defined by users based on their experiences and are not universal.

This paper begins by using the definition of storm surge to calculate the threshold. It is widely known that typhoon-storm surge refers to abnormal rises and falls in sea levels caused by weather factors, indicating that when a storm-surge event occurs, the residual water levels calculated by the tide-gauge data can be regarded as outliers. Therefore, this paper attempts to calculate the threshold based on detecting the outlier. Four commonly used methods for detecting the outlier—the Pauta criterion (PC) [13], Chauvenet criterion (CC) [14,15] Pareto distribution (PD) [16] and kurtosis coefficient (KC) [17]—were used to calculate the annual threshold criteria for each tide-gauge station for outlier detection, after which the storm surge events at each station were detected. Li et al. [18] used the Pauta criterion to detect outliers in groundwater time-series data and obtained ideal results by correcting outliers. Luo et al. [19] used the Pauta criterion, Chauvenet criterion and Fang Guohong criterion to detect the outliers in tide-level data measured at offshore tide-gauge stations and conducted a harmonic analysis on the measured tide-level data after the discrimination. The results showed that the Pauta criterion can better discriminate the dense and continuous occurrence of outliers in tide-level data series. Xiong et al. [20] analyzed the characteristics of four outlier-detection methods based on the Pauta criterion, Grubbs criterion, Chauvenet criterion and Dixon criterion [21]; their results showed that when the number of samples exceeds 185, the Pauta criterion is more accurate for the detection of outliers. Wang et al. [22], who compared the outlier-detection capabilities of the Pauta criterion and Chauvenet criterion in wavefront data with a sample number of 9500, considered that the use of the Chauvenet criterion can prevent the deletion non-abnormal data and that it is better in the testing of wave-surface data. The Pareto distribution is commonly used in modeling extreme events and can be used to model the distribution of storm surges [23,24,25,26]. For instance, researchers can use the Pareto distribution to model the maximum storm surges at a specific location and use the model to detect extreme events based on a given threshold. The kurtosis coefficient (KC) is an indicator for measuring outlier degree of outlier data [27,28]. In the context of storm surges, it can be used to determine the shape of sea-level data and differentiate between normal and extreme events. The higher the kurtosis, the greater the peak in the distribution, indicating the presence of extreme values or outliers. This information can be used to detect storm surges and distinguish them from normal sea-level fluctuations.

These four methods have been used to detect outliers in oceanographic data and have achieved good results. This paper first reports the application of the four methods to the detection of typhoon-storm-surge events. Each discrete set of outliers is a storm-surge event. Subsequently, the storm-surge events detected by the four methods are obtained by using the storm-surge-event-recognition algorithm to identify and count the previously detected discrete storm-surge events (note: a typhoon-storm-surge event includes information such as the time of the typhoon and the scope of influence). Combined with the evaluation indicators of the deep-learning target-detection algorithm and the historical typhoon-storm-surge events recorded in reference [29], *Bulletin of China Marine Disaster* (these materials are available at https://www.mnr.gov.cn/sj/sjfw/hy/gbgg/zghyzhgb/index_1.html (accessed on 20 December 2022)) and landfall-tropical-cyclone data [30,31], the four methods are evaluated to determine which is the most reliable for detecting typhoon-storm-surge events along the coast of China. We hope to lay a foundation for follow-up quantitative research on the intensity of storm-surge events along the southeast coast of China, as well as research on the evolution of the climate scale and the characteristics of temporal and spatial variability.

## 2. Materials and Methods

This research mainly uses three types of data: long-time-series hourly-tide-level data along the coasts of China and residual-water-level data acquired by harmonic analysis of hourly-tide-level data; six-hour sea-level-pressure data used in the inverse air-pressure correction for the residual-water-level data; and historical typhoon-storm-surge-statistics data for comparative validation. The specific data are described below.

### 2.1. Data

#### 2.1.1. Tide Gauge (TG) Data

Hourly sea-level data from 17 tide gauges on the coasts of China were used (Figure 1a). The data were obtained from the University of Hawaii Sea Level Center (UHSLC) (data are available at https://uhslc.soest.hawaii.edu/data/ (accessed on 20 December 2022)). The sea-level data span the period 1954–2018. From the 17 records available, 14 are longer than 23 years (Figure 1b). Two pairs of records were combined and the data records of Macau station with the fewest records were excluded, leaving 14 records covering at least 18 years for the analysis.

The nearby Dalian and Laohutan stations have data for the periods 1975–1990 and 1991–1997, respectively. However, at Dalian, the tide gauge was relocated twice, once at the end of 1976 and once in 1979. After 1979, it shared the same position as Laohutan. Thus, the data for the periods 1980–1990 at Dalian and 1991–1997 at Laohutan were combined.

In Hong Kong, the tide gauge was located at North Point for the period 1962–1986, and then moved half a kilometer away to Quarry Bay for the period 1987–2012. The same data applied to both locations with an offset of 1.02 cm [32]. Thus, the two Hong Kong records were merged into one 57-year record, the longest in the study. These tide-gauge records are from newer instruments and do not contain the types of error described by Lentz et al. [33] and Harms et al. [34].

The quality control performed included visual checks of the time series of the hourly measurements and the tidal residuals. Checks were also performed on the estimated annual tidal constants at each station. Values with spurious jumps, data shifts and time shifts from the values in other years were excluded. If more than 30% of the values were missing in a year, the data for that whole year were excluded. Using this quality-control method, data for 1988 at Lusi, 1996 at Shanwei, 2013 at Kaohsiung and 1998–1999, 2001–2003 and 2005 at Keelung were removed. The periods with valid observations are shown in Figure 1b.

#### 2.1.2. NCEP-Reanalysis Air-Pressure Data

The sea-level-pressure data used for reverse air-pressure correction of tide-gauge-station data are from the Reanalysis I Project of National Centers for Environmental Prediction/National Center for Atmospheric Research (NCEP/NCAR). The project assimilates past data from 1948 to the present by using the analysis/forecast system [35] (data are available at http://www.esrl.noaa.gov/psd/data/gridded/data.ncep.reanalysis.surace.html (accessed on 20 December 2022)). The time span of gridded sea-level pressure (SLP) data from the NCEP reanalysis data was from January 1948 to the present, with a temporal resolution of 6 h and a global latitude/longitude resolution of 2.5°. In this paper, the six-hour sea-level-pressure data during the recording of the tide-gauge-station data were linearly interpolated to correspond to the tide-level data of the hourly tide-gauge station above.

#### 2.1.3. Historical Typhoon-Storm-Surge Events (Historical Statistics)

The historical typhoon-storm-surge events (also called historical statistics) were mainly from *Bulletin of China Marine Disaster*, landfall-tropical-cyclone data, typhoon-track information and the reference [29]. The *Bulletin of China Marine Disaster* only records the number of typhoon-storm-surge events that occur each year (without recording the time of occurrence and the scope of influence) and a detailed description of a few typical storm surges (causing serious disasters). It is available on the website of Ministry of Natural Resources of the People’s Republic of China (https://www.mnr.gov.cn/sj/sjfw/hy/gbgg/zghyzhgb/index_1.html (accessed on 20 December 2022)), with a time span of 1989–2021. The landfall-tropical-cyclone data are available at China Meteorological Administration (CMA) Tropical Cyclone Data Center (https://tcdata.typhoon.org.cn/en/index.html (accessed on 20 December 2022)). The data provide information on typhoons that have occurred in China since 1949, including landfall time and location and typhoon number. It is worth noting that typhoons that do not occur in China can also cause typhoon-storm surges (for example, Typhoon No.0014, SaoMai, which did not occur in China but tracked along the coastline, caused a relatively severe typhoon storm surge and caused serious disasters [29]); therefore, typhoon-track information is complementary to CMA data. The typhoon-track information is available at: https://typhoon.slt.zj.gov.cn/default.aspx (accessed on 20 December 2022). The landfall-tropical-cyclone data and the typhoon-track information can provide information that is not recorded in the *Bulletin of China Marine Disaster*, such as the time of the typhoon’s occurrence and the scope of its influence. Historical statistics can be obtained by combining the three kinds of data above, including the annual frequency of typhoon-storm surges, the time of occurrence of each storm surge and the scope of its influence.

*Collection of Storm Surge Disasters Historical Data in China 1949–2009* is a book written by Yu et al. [29]. The book collects and organizes the processes of 209 typical (not all) storm surges along the coasts of China from 1949 to 2009 and records relevant typhoon information including typhoon number, start and end time and influence range, as well as the disaster influence, residual water levels and high tide levels beyond the local-warning tide level of each storm-surge process. According to the relationship between the maximum tide level of each influenced tide gauge and the local-warning tide level in a storm-surge process, the book divides storm surges into four levels: red, orange, yellow and blue. This paper collects the relevant information from 84 typical typhoon storm surge events recorded from 1975 to 1997, including 15 red-level storm-surge events, 36 orange-level storm-surge events, 27 yellow-level storm-surge events and 6 blue-level storm-surge events. These types of data are mainly used in Section 3.3.

It should be noted that the historical typhoon-storm-surge events recorded in the above materials are all blue-level and above. The blue level indicates a storm surge when one or more representative tide gauge in the affected area along the coast has a high-tide level that is 30 cm lower than the local-warning tide level (https://www.mnr.gov.cn/gk/bzgf/201004/t20100402_1971711.html (accessed on 20 December 2022)).

### 2.2. Method

This paper reports the acquisition of the residual-water-level data by carrying out missing-value interpolation, harmonic analysis, reverse air-pressure correction and other pre-processing methods for obtaining hourly tide-level data of tide-gauge stations. Figure 2 shows the relationship between the water level, the astronomical tide and the residuals. In addition, it presents the calculation of the thresholds of the annual residual-water-level data of the above tide-gauge stations by using PC, CC, PD, KC (Figure 2) methods, detection of the storm-surge events and screening using the storm-surge-event-recognition algorithm to determine storm-surge events. The specific method is described as follows.

#### 2.2.1. Data Pre-Processing of Tide-Gauge Stations

For the years in which the proportion of the missing data of tide gauge stations is less than 30%, the evaluation of artificial circular-approach method [36] was first used for interpolation processing t; next, the S_TIDE [37,38] program package was used to carry out a harmonic analysis of the tide-gauge stations on an annual basis to acquire initial residual-water-level data; finally, the inverse barometer (IB) correction [39,40,41] of the residual-water-level data was carried out to acquire the final residual-water-level data. A 12-hour Gaussian filter (Figure 3) was used for the low-pass filtering of the final residual-water-level data. Considering that the low-pass filter removed high-frequency signals, dynamic atmospheric correction was not used [42].

#### 2.2.2. Storm-Surge-Event Detection and Identification Method

For the above residual-water-level data obtained after quality inspection and reverse-air-pressure correction, a threshold was obtained by using the four methods below. The outlier sequences in the residual-water-level data sequence of each tide-gauge station were detected according to the threshold and the final storm-surge events of each tide-gauge station were acquired by screening based on the storm-surge-event-recognition algorithm.

Pauta Criterion

Pauta criterion (PC) [18] is based on the assumption that samples obey a normal distribution, arguing that when the absolute value of the difference between the detected value and the mean value exceeds the standard deviation of the samples by three times, the detected-value data are abnormal and the abnormal data and the corresponding dates are output.

Taking the hourly residual-water-level data H within 1 year as the sample data and assuming that the residual-water-level data obey a normal distribution, the method for calculating the mean value $\bar{H}$ and standard deviation S of the samples is shown in Equations (1) and (2):(1)$$\bar{H}=\frac{\sum_{i=1}^{N}H_{i}}{N}$$
(2)$$\;S=\sqrt{\frac{\sum_{i=1}^{N}{\left(H_{i}-\bar{H}\right)}^{2}}{N-1}}$$
(3)$$\left|H_{i}-\bar{H}\right|>3S,\;\mathrm{and}\;H_{i}>0$$
where$\;H_{i}$ is the residual-water-level data at a certain time. When the difference between $\;H_{i}\;\mathrm{and}\;\bar{H}$ exceeds ±3S, $\;H_{i}>0$ is considered an outlier, that is, $H_{i}$ is a possible storm-surge event, with threshold of$\;u=\bar{H}+3S$.

2.Chauvenet Criterion

Chauvenet criterion (CC) [14,15] assumes that the samples obey the normal distribution, arguing that, out of *N* data points, the data points with a probability of occurrence of less than 1/2 *N* can be considered as outliers and determined as possible storm-surge events. Let Z be a value greater than 0; when $H_{i}-\bar{H}\leq ZS$, the following function relationship is satisfied:(4)$$P\left(\frac{\left|H_{i}-\bar{H}\right|}{S}\leq Z\right)=2\frac{1}{\sqrt{2\pi }}\int_{0}^{Z}e^{-\frac{t^{2}}{2}}dt$$
where $H_{i}$ is the residual-water-level data at a certain time, $\bar{H}$ is the mean value of the residual water level within 1 year and *N* is the number of data points within 1 year. Assuming that there is a certain $Z_{c}$, called the Chauvenet coefficient, when $\frac{\left|H_{i}-\bar{H}\right|}{S}>Z_{c}$, it is considered that data points with a probability of less than 1/2 *N* exist, then:(5)$$P\left(\frac{\left|H_{i}-\bar{H}\right|}{S}>Z\right)=\frac{1}{2N}$$

By combining Equations (4) and (5), the following equation can be obtained:(6)$$2\frac{1}{\sqrt{2\pi }}\int_{0}^{Z}e^{-\frac{t^{2}}{2}}dt=1-\frac{1}{2N}$$

When *N* is known, $Z_{c}$ can be solved according to Equation (6). If $\left|H_{i}-\bar{H}\right|>Z_{c}S,\;\mathrm{and}\;H_{i}>0$, $H_{i}$ can be considered an outlier, that is, $\;H_{i}$ is the possible storm-surge event, where the threshold $u=\bar{H}+Z_{c}S$.

3.Pareto Distribution

Pareto distribution (PD) [16,23,24,25,26], which is named after the Italian civil engineer, economist and sociologist Vilfredo Pareto, is a power-law distribution found in a large number of real-world phenomena. Beyond economics, PD is also known as Bradford distribution.

If X is a random variable that obeys the PD (Pareto I), then X is greater than the probability distribution of a certain value $x_{\;}$, that is, survival function, as shown in Equation (7):(7)$$F_{X}\left(x\right)=P_{r}\left(X>x\right)=\left\{\begin{array}{c} {\left(\frac{x_{\;m}}{x}\right)}^{\alpha }\;x\geq x_{m},\;\\ \;1\;x<x_{m}. \end{array}\right.$$
where $x_{\;m}$ is the minimum possible value of X ($x_{\;m}$ is a positive value) and α is a positive parameter. Pareto I distribution is characterized by a scale parameter $x_{\;m}$ and a shape parameter α. When the distribution is used to simulate wealth distribution, parameter α is called the Pareto index. Its distribution-density function is as follows:(8)$$f_{X}\left(x\right)=\left\{\begin{array}{c} \frac{\alpha X_{m}^{\alpha }}{x^{\alpha +1}}\;x\geq x_{m},\;\\ \;\;0\;x<x_{m}. \end{array}\right.$$

In this paper, the maximum-likelihood function is used to estimate scale parameter $x_{\;m}$ and shape parameter α. The parameter estimation obtained by maximum-likelihood-estimation method has asymptotic unbiasedness, effectiveness, consistency and other good statistical properties. The maximum-likelihood estimation of $x_{\;m}\;\mathrm{and}\;\alpha$ is:(9)$$\;{\hat{x}}_{m}=\underset{i}{min\;}x_{i}.$$
(10)α^=nΣilnxix^m.
where $P_{r}$ is the ratio of the residual-water-level-data duration during annual storm surge to the residual-water-level-data duration for the whole year. Based on the above equation, each parameter was obtained by using MATLAB programming and the threshold u=x was obtained, after which the potential storm-surge events were detected.

4.Kurtosis Coefficient

The kurtosis coefficient (KC) [17] is an indicator for measuring outlier degree of outlier data. The greater the KC, the more the extreme values in the data series. The specific procedure is as follows:

Step 1: Calculate the KC of the sample data:(11)$$K_{n}=\frac{\frac{1}{n}\sum_{i=1}^{n}{\left(H_{i}-\bar{H}\right)}^{4}}{{\left(S_{n}^{2}\right)}^{2}}$$
where $\bar{H}$ and $S_{n}^{2}$ represent the mean value and variance of the sample, respectively:(12)$$\bar{H}=\frac{\sum_{i=1}^{N}H_{i}}{N}$$
(13)$$\;S_{n}^{2}=\frac{1}{N-1}\overset{N}{\underset{i=1}{\sum }}{\left(H_{i}-\bar{H}\right)}^{2}$$

Step 2: After KC $K_{n}$ is calculated by the above equation, if $K_{n}\geq 3$, then the data $H_{i}$ that maximize ${\left(H_{i}-\bar{H}\right)}^{2}$ value are removed from the samples. Next, repeat the two steps above until $K_{n}<3$. The maximum value $H_{i}$ is selected out of the remaining sample data as the threshold u for detecting storm-surge events u.

5.Storm-Surge-Event-Identification Algorithm

There are two conditions that can influence the detection of storm-surge events. One is errors in the data, and the other is shallow-water effects. There are two basic types of error in data: one is that outliers are significantly higher than adjacent values; the other is the phase shift, which produces a large number of tidal-like fluctuation residuals [43,44]. However, these errors can be detected easily from the data residuals after tidal components are removed from water-level data. Outliers appear as spikes, completely different from storm surges. Incorrect phase shifts can be easily identified and corrected from the fluctuation pattern of the residuals. At a harbor gauge, water-level data can be affected by strong shallow-water effects. The shallow-water effects can be removed by filtering the residuals using a low-pass (Gaussian) filter. Numerical tests (Figure 3) showed that a 12-hour filter can reduce shallow-water effects. However, this low-pass-filter also suppresses the amplitude of storm surges with a duration of less than 12 h. To reduce shallow-water effects and maintain the original storm-surge information, three steps were taken to identify a storm event. First, possible storm-surge events were identified from the residuals using the threshold criterion described above; next, storm-surge events were also identified from the low-pass-filtered residuals; finally, only the storms detected in both unfiltered residuals and low-pass-filtered residuals were retained.

In some cases, it was found that two storm-surge events occurred very close in time (they may have belonged to the same storm-surge event) when checking the storm surge events detected by the four methods according to the threshold, which may have influenced statistics of storm-surge events. This may have been because the water level continued to oscillate after the center of the typhoon passed due to free movement of the water in returning to its normal level and changes in wind direction [45]. These oscillations may have been incorrectly counted as new storm-surge events. However, they were generally smaller than one tidal cycle. Therefore, a tidal cycle (12 h for the semidiurnal tide along the coast of China) was chosen as the criterion for identifying storm-surge events. If the interval between two storm-surge events is more than 12 h, they are considered to belong to different storm events; otherwise, they are regarded as the continuation of a single storm-surge event.

Since this paper mainly studies typhoon-storm surges, it is necessary to identify the storm surges that occurred from May to November from the storm events identified above. This is because the typhoon-storm surges along the coasts of China mainly occurred in summer and autumn (from May to November).

#### 2.2.3. Comprehensive Evaluation Indicators

In order to compare the four methods, the evaluation indicators of the deep-learning target-detection algorithm [46] were introduced to evaluate comprehensive detection capabilities of the four detection methods.

Precision refers to the ratio of the number of typhoon-storm-surge events detected by a certain detection method (that is, the number that matches the historical typhoon-storm-surge events) and to the total number of storm-surge events detected by this method. Recall refers to the ratio of the number of typhoon-storm-surge events detected by a certain detection method (that is, the number that matches the historical typhoon storm surge events) and to the total number of historical typhoon-storm-surge events. The P and R indicators sometimes contradict each other; therefore, they need to be considered comprehensively. The most common method is F-Measure (also known as F-Score). The F-Measure is a harmonic mean value of precision and recall, as well as an evaluation indicator that integrates the two indicators. The F-Measure comprehensively reflects the overall indicator. The higher the F-Measure, the better the comprehensive detection effect of the method, which can best meet the needs of research.

F-Measure is the weighted harmonic mean value of Precision and Recall.
(14)$$F=\frac{\left(\alpha^{2}+1\right)P\times R}{\alpha^{2}\left(P+R\right)}$$

When α = 1, it is the most common F1, that is,
(15)$$F=\frac{2\times P\times R}{P+R}$$

According to Equation (15), F1 combines the results of P and R. A high F1 this can indicate that the detection method has a better effect.

## 3. Results

### 3.1. Feasibility Analysis of the Four Methods

This section verifies the feasibility of applying the four methods in the detection of storm-surge events by analyzing the thresholds calculated by the methods and the spatial distribution of the detected storm-surge events. The thresholds were calculated by using PC, CC, PD and KC method at each tide gauge station and the statistics of the detected storm surges are shown in Table 1 and Figure 4.

In order to make the detection results of each station comparable, we only compare the detection results of all the tide-gauge stations during the overlapping years (1980–1996). According to Table 1 and Figure 4, the threshold calculated by the CC was the largest out of the four methods and the number of detected storm-surge events was the lowest with this method. At most of the stations, the threshold calculated by the KC was the lowest and the number of the detected storm-surge events was the largest with this method. Clearly, the detected storm events were negatively correlated with the threshold value.

As shown in Table 1, the thresholds of the tide gauges (Lusi and Lianyungang) along the coasts of the Bohai Sea and the Yellow Sea calculated by the four methods were greater than those of the other stations. For example, the three stations with the highest thresholds based on the PC and CC were Lusi, Dalian and Lianyungang, respectively, while the three stations with the highest thresholds based on the PD and KC were Xiamen, Lianyungang and Lusi, respectively. On the whole, the thresholds calculated by the four methods showed that the threshold of the northern coast was higher than that of the southeastern coast, which may be because the northern coastal sites were both affected by typhoon-storm and extratropical-storm surges.

The storm-surge events detected by KC occurred more frequently along the coasts of the Bohai and Yellow Seas (Table 1). The storm-surge events detected by the three methods other than the KC occurred with a high frequency along the southeastern coast of China. Another interesting phenomenon is that the three methods of PC, CC and KC detected the highest frequency of storm-surge events at Lusi Station in Jiangsu Province. According to the statistics, more than half of the storm surge events at Lusi Station occurred in May, October and November (PC (22/40); CC (9/17); PD (15/29); KC (51/88)), which is a period in which extratropical storm surges frequently occur [47]. From the spatial distribution of the detected storm-surge events above, it can be seen that all four methods are feasible for detecting storm-surge events.

### 3.2. Capability Evaluation of the Four Methods for Detecting Storm-Surge Events

#### 3.2.1. Preliminary Capability Evaluation of the Four Methods for Detecting Storm-Surge Events

This section offers an initial demonstration of the ability of the four methods to detect storm surges. As is well known, the range of influence of storm surges is generally between tens of kilometers and thousands of kilometers and the duration varies from a few hours to hundreds of hours. Therefore, a storm surge may influence multiple tide-gauge stations. In order to count the annual storm-surge events detected by each method, this section summarizes and counts the storm-surge events detected by each station according to time and location (in time, the time is close and continuous and, if there is an interval, this interval cannot exceed 24 h; in space, the site location is adjacent in space and the distance does not exceed 500 km [6,12]) to acquire the number of storm surges detected by the four methods each year, as shown in Figure 5.

It is worth noting that this section only counts storm surges that occurred from 1975 to 1997, since most of the TG data cover the dates of 1975–1997 (Figure 1b). The historical typhoon-storm-surge events in Figure 5 were statistically obtained from *Bulletin of China Marine Disaster*, the landfall-tropical-cyclone data [30,31] and typhoon-track information materials.

Figure 5 shows that the CC detected the fewest storm-surge events in each year, whereas the KC detected the highest number of storm surges in most of the years. The annual increase and decrease trend of the number of storm-surge events detected by the four methods was roughly the same (except for the CC in 1984, which showed a relatively obvious and completely opposite trend). The number of storm-surge events detected by the KC in 1992 was abnormally high compared with the historical typhoon -storm-surge-event number (eight deviations).

Comparing the storm-surge events detected by the four methods with the historical statistical data, we found that some of the detected storm surges were not consistent with the historical statistical data, which clearly showed that there were errors in the detection results. For example, in 1989, there were 15 typhoon-storm surges in the historical statistics and the KC also detected 15 storm-surge events; however, by comparing the storm-surge information (the occurrence time and affected stations) from the two, it was found that 1ten of the storm-surge events detected by the KC were consistent with the historical statistical data and the other five were not typhoon-storm surges. By checking the characteristics of these error events (occurring in the spring and autumn, the stations are located in the northern coastal areas and the tide surge process is relatively gentle), it can be judged that these error events were extratropical-storm surges. In order to initially compare the detection capabilities of the four methods for typhoon-storm surges, it was necessary to determine the error events detected by the four methods. In this paper, by combining landfall-tropical-cyclone data [30,31] and typhoon-track information, the detected storm -surge events that did not occur during a typhoon and took place far away from the typhoon tracks were identified as error events. Figure 6 shows the comparison histogram from 1975 to 1997 between the number of storm-surge events detected by the four methods (shown in green bar and labeled ’Total number of detections’) the number of storm-surge events identified as error events (shown in blue bars and labeled ‘Numbers of False Detections’) and the number of historical storm-surge events (shown in the red bar and labeled ‘True number in history’).

This paper introduces two parameters, the maximum false-detection rate (MFDR) and the overall false-detection rate (OFDR), to preliminarily evaluate the detection capabilities of the four methods. The MFDR, the ratio of the maximum value of the NFDs to the ‘Total number of detections,’ can reflect the upper limit of the error for a certain method. The OFDR is the ratio of the sum of the NFDs of all the years to the sum of the ‘Total number of detections’ for all the years, which can reflect the accuracy of a particular method. The MFDR and OFDR results (highlighted in yellow) of the four methods are shown in Figure 6. It was found that the MFDR and OFDR of the KC were the highest, indicating that the KC may cause greater errors in the detection results; furthermore, the fact that the KC had the highest overall false-detection rate indicates that its accuracy was the lowest. On the other hand, the MFDR and OFDR of the CC were the lowest, indicating that the CC had the smallest error in the detection results and the highest accuracy. Note that the number of error events was proportional to the amount of extra work.

#### 3.2.2. Comprehensive Capability Evaluation of the Four Methods for Detecting the Storm-Surge Events

To compare the comprehensive detection capability of the four methods, this paper introduces the evaluation indicators of deep-learning-target-detection algorithms [46]. Table 2 shows the calculation results of the evaluation indicators of the four methods (see Section 2.2.3 for details).

The P represents precision, R represents recall and F1 is a harmonic mean value of P and R, as an evaluation indicator that integrates the two indicators. According to the annual calculation results in Table 2, the year with the highest PC precision (P = 1) was 1991, indicating that the number of typhoon-storm-surge events detected in that year was consistent with the historical statistics. The year with the lowest PC precision (P = 0.5) was 1977, which can also be concluded from Figure 6 (Section 3.2.1, MFDR = 0.5 also occurred in this year). The year with the highest PC recall (R = 0.88) was 1983, indicating that the number of typhoon-storm-surge events detected in 1983 was the closest to the total number of historical typhoon-storm-surge events. The year with the highest comprehensive evaluation indicator (F1 = 0.82) for the PC was 1983, which indicates that the detection result of the PC in 1983 both ensured the precision and was the closest to the number of historical storm-surge events in that year. The results of the other three methods are shown in Table 2 and are not repeated here. Regarding the overall results, the CC had the highest precision (consistent with the results acquired in Section 3.2.1), but the lowest recall. By contrast, the KC had the lowest precision but the highest recall. It can be seen that, sometimes, there was a contradiction between the precision and the recall. In these cases, it was necessary to use the comprehensive evaluation indicator to evaluate the overall detection capability of the method. According to the F1 results of the four methods, the PC had the best comprehensive detection capability for the typhoon-storm-surge events, followed by the KC, while the CC was the lowest.

### 3.3. Detection Results of Typical Storm Surges Detected by the Four Methods

Table 3 shows the comparison between the typhoon-storm-surge events detected by the four methods and the typical storm-surge events in the reference [29] from 1975 to 1997. This section can be used to verify the conclusion of Section 3.2.

For the red-level storm-surge events, the detection rates of the PC and KC were as high as 100%; for the orange- and blue-level storm-surge events, the detection rates of the PC were higher than or equal to those of the other methods; for the yellow-level storm-surge events, the detection rates of the PC were slightly lower than those of the PD. Comparing the overall detection results for the typical storm-surge events detected by the four methods, it can be concluded that the PC (detection rate 0.90) was significantly stronger than the other three methods, followed by the KC, while the CC was the worst.

## 4. Discussion

It is feasible to use PC, CC, PD and KC outlier-detection methods to detect typhoon-storm surges along the coast of China. The detection results were analyzed and compared with the historical typhoon storm surge events. Detailed discussions are presented below.

### 4.1. The Detection Results of CC and PD Were Less Influenced by the Extratropical-Storm Surges

From the threshold values calculated by the four methods at each TG station from 1980 to 1996 (Table 1, Figure 4), it can be seen that the four methods obtained different threshold values for the same TG station. Each discriminant criterion has its own processing method, which leads to inconsistent discrimination between outlier (hereafter, outlier refers to storm-surge events) results with different criteria. The threshold calculated by the CC was the highest and the detected outliers (storm-surge events) were the lowest with this method. According to Xiong et al. [20] and Wang et al. [22], it is known that this relates to the number of samples. Their research found that if the number of samples *N* < 185, the threshold calculated by the CC is more appropriate than the PC for detecting outliers; if the number of samples *N* > 185 (N = 8760), the ratio of the threshold calculated by the CC to the standard deviation is >3 (that is, the Chauvenet coefficient Zc > 3), whereas the ratio calculated by the PC and KC is ≤3. That is, when the number of samples is large, the threshold calculated by the CC is higher than those of the PC and KC. This can lead to the detection of outliers with large errors, whereas smaller outliers are often missed.

According to the threshold-calculation formulas of the four methods (Section 2.2.2), the thresholds calculated by the PC, CC and KC are related to the mean and standard deviation. Based on the overall results, the threshold calculated by the four methods in the northern coastal areas (Bohai Coastal Station and Yellow Sea) was high, whereas the threshold in the southern coastal areas was low. The northern coastal areas of China are mainly affected by extratropical storm surges and typhoon-storm surges occasionally occur. Extratropical-storm surges can occur all year round, with the highest frequency in spring and autumn; their average annual frequency is higher than that of typhoon-storm surges [48]. In this case, there is a large number of outliers with small magnitudes (relative to the magnitudes of the outliers caused by typhoons), resulting in a higher threshold. A high threshold may cause the missed detection of moderate storm-surge events, but, at the same time, extratropical-storm-surge events with small water increases can be eliminated. These conclusions show that the four methods responded to the influence of extratropical-storm surges (the threshold is larger than when only a typhoon-storm-surge occurs), which is beneficial for reducing the impact of extratropical-storm surges when detecting typhoon-storm surges. The CC and PD elicited larger responses, which indicates that their detection results were less affected by the extratropical-storm surges.

The analysis of the number of storm-surge events detected by each tide gauge shows that the frequency of typhoon-storm surges on the southeast coast of China was higher than that on the north coast. At Lusi station, the three methods of PC, CC and KC detected the most storm-surge events. The reason for this may be that Lusi station is located in the junction of the northern and southern coastal areas and is simultaneously affected by typhoon-storm surges and extratropical-storm surges [47].

Overall, the spatial distribution of the frequency of the storm surges detected by the four methods was consistent with the results of previous studies [49], which shows that these four methods are feasible for detecting storm-surge events along the coast of China. For the regions affected by both extratropical-storm surges and typhoon-storm surges, the CC and PD can better reduce the impact of extratropical-storm surges than the PC and KC when detecting typhoon-storm surges. The storm-surge-threshold criterion calculated by the CC is much higher than those of the other three methods, indicating that the CC is more focused on detecting the largest, most damaging storms. When typhoon-track information cannot be used to exclude the influence of extratropical-storm surges, it is recommended to use the CC and PD to detect typhoon-storm surges.

### 4.2. Evaluation of the Four Detection Methods

#### 4.2.1. Analysis of False-Detection Results

A storm surge may have a wide range of influences and may be detected by multiple tide gauges at the same time. Therefore, it was necessary to combine the annual detection results of storm-surge events at each tide gauge according to the occurrence time, typhoon path and other information, followed by an examination of the statistics on the annual number of storm -urge events detected by the four methods during 1975–1997. By comparing the annual results from the four methods with the number of historical annual typhoon-storm surges, the detection errors for each method were obtained (Figure 6). These error events may have been extratropical-storm-surge events. The results in Figure 6 show that the CC had the lowest MFDR, which indicates that most of the storm-surge events detected by the CC were consistent with the storm-surge events that occurred over time. Because of the 12-hour low-pass (Gaussian) filter and excluding the extratropical-storm surges with gentle storm-surge processes and low residual-water levels by virtue of the high threshold calculated by the CC, most of the storm surges detected by the CC were made to seem more severe. This conclusion is consistent with that in Section 3.1. The KC had the highest MFDR, indicating that more extratropical storm surges may have been falsely detected by the kurtosis method. The greater the number of error events, the lower the detection accuracy of this method. This indicates that the KC has the lowest accuracy, whereas the CC has the highest accuracy.

#### 4.2.2. Comprehensive Capability Evaluation

The results calculated by the target-detection-evaluation indicators showed that the CC had the highest precision, which was consistent with the analysis of the false-detection results (Section 3.1). The CC had the lowest recall of the four methods, indicating that the number of typhoon-storm-surge events detected by the CC was far lower than the historical-statistics data. The precision of the KC was the lowest, indicating that the typhoon-storm-surge events detected by the KC accounted for a low proportion of the total detection results by the KC and that extratropical storm surge events may have contributed to the remaining proportion (this conclusion is consistent with the previous section). However, the KC had the highest recall, indicating that the typhoon-storm-surge events detected by the KC accounted for the highest proportion of the corresponding historical-statistics data. In order to comprehensively consider the precision and the recall, the harmonic-average F1-Measure of the precision and the recall was used as a comprehensive evaluation indicator. The higher the F1-Measure, the better the comprehensive detection effect of the method, which indicated that the method can satisfy the needs of research. Table 2 shows the calculation results of the F1-Measure. The fact that the PC had the highest F1-Measure (F1 = 0.66) indicates that its comprehensive detection ability was the best. The fact that the CC had the worst comprehensive detection ability (F1 = 0.57) was mainly due to the fact that it had the lowest recall (R = 0.42, which was much lower than the recall of the other three methods, all of which were higher than 0.6).

#### 4.2.3. Detection Capability of Typical Storm Surge

This paper collected and organized all the typical storm-surge events, which caused different degrees of disaster and loss in coastal areas, from 1975 to 1997 from the reference [29]. Comparing the detection results of the four methods with the statistical results of the typical storm-surge events above (Table 3), it was found that the PC had the highest detection precision for both the graded and all the typical storm-surge events, except for the yellow level, which was slightly below that obtained with the PD. The performance of the KC was second only to that of the PC. However, the CC and PD had different degrees of missed detection of red-level storm surges, which is a very serious error. A red-level storm surge can cause so much damage that it must be detected by detection methods.

Based on the above discussions, we suggest that the appropriate method can be selected according to the research preferences: if the research purpose is to detect as many typhoon-storm-surge events as possible (the highest recall), it is recommended to use the KC method. However, the disadvantages of the KC must be taken into account; that is, more work is required to eliminate the large number of errors in the KC detection results (the highest OFDR) by collecting auxiliary data, such as typhoon-track information. If the research purpose is to analyze some extremely severe typhoon-storm-surge events (the highest precision) from history, it is recommended to use the CC. If the research purpose requires the best comprehensive detection ability (the highest F1), it is recommended to use the PC for detection.

## 5. Conclusions

Based on the hourly residual-water-level data from 14 tide-gauge stations along the coasts of China, four outlier-detection methods, PC, CC, PD and KC, were applied to detect typhoon-storm-surge events. The historical typhoon-storm-surge events from 1975 to 1997 were counted as comparative verification data. The evaluation indicators of the deep-learning target-detection algorithm were introduced to comprehensively evaluate the detection abilities of the four methods. Finally, we arrived at the conclusions described below.

The PC has the best comprehensive detection ability for typhoon-storm-surge events (the highest F1), making it the most suitable for typhoon-storm-surge detection in coastal areas of China. The CC has the highest detection precision for the largest, most damaging typhoon-storm surges (the highest precision) and the lowest OFDR (overall false-detection rate), which indicates that if the focus is on severe typhoon-storm surges, CC is the most suitable method. The KC can detect the most typhoon-storm-surge events (the highest recall), but it also introduces a large number of error events. These error events need to be eliminated by collecting auxiliary data, such as typhoon-track information, which involves a huge workload.

In the context of global climate change, it is not clear whether there will be new changes in occurrence frequency, surge height, influence range or geographical distribution patterns in future storm-surge events. In future research, we will consider improving the accuracy of storm-surge-event detection by optimizing or improving the algorithm, such as by adding parameterized typhoon-track information to the algorithm so as to study the long-term temporal and spatial variation in storm-surge activity along the coasts of China.

## Figures and Tables

**Figure 1 ijerph-20-03253-f001:**
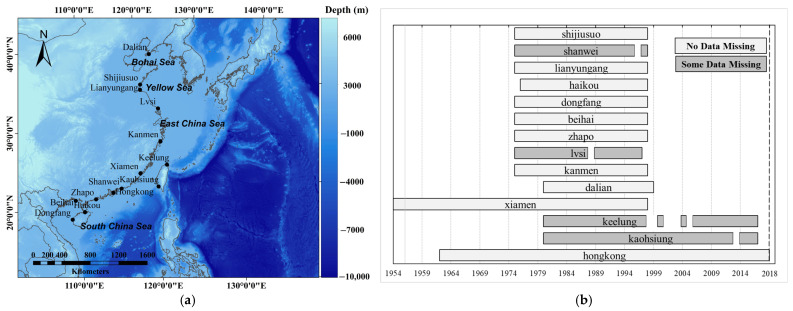
(**a**) Locations of tide-gauge stations along the southeast coast of China and (**b**) Gantt chart for the corresponding periods of valid data. The bathymetry in the studied areas is also shown in Figure 1a, which is based on the GEBCO_2021 Gridded Bathymetry Dataset from the GEBCO (The General b=Bathymetric Chart of the Oceans, https://www.gebco.net/data_and_products/gridded_bathymetry_data/ (accessed on 20 December 2022)) website.

**Figure 2 ijerph-20-03253-f002:**
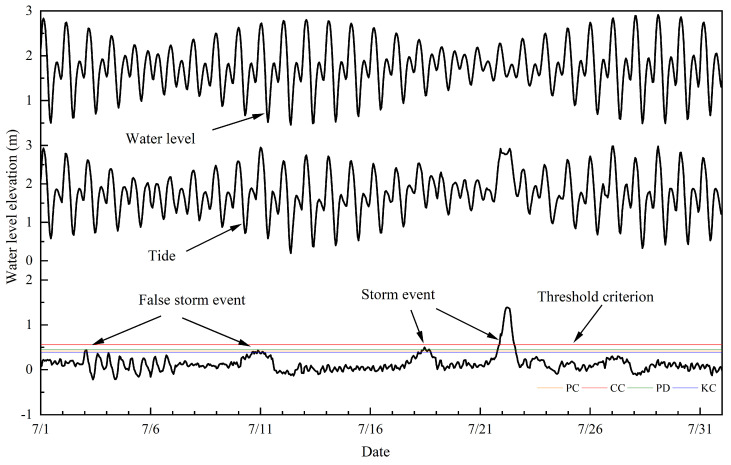
Water level, tide, and storm surge at Zhapo tide gauge during July 1980. Color lines indicate the thresholds calculated by the four methods. Orange line indicates PC; red line indicates CC; green line indicates PD; blue line indicates KC. Note: PC is an abbreviation for Pauta criterion; CC is an abbreviation for Chauvenet criterion; PD is an abbreviation for Pareto distribution; KC is an abbreviation for kurtosis coefficient.

**Figure 3 ijerph-20-03253-f003:**
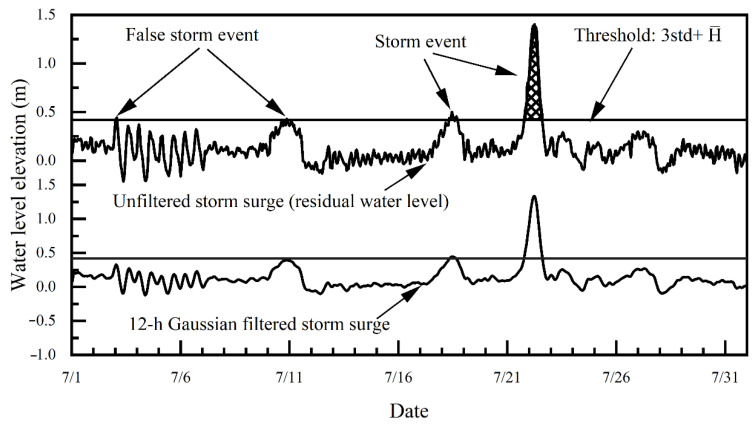
Storm surge and filtered storm surge at Zhapo tide gauge during July 1980. Note that the shallow-water effects are removed by the 12-hour Gaussian filter, but the surge amplitude of storm events with a duration of less than 12 h is only damped by the low-pass filter. Threshold is calculated by Pauta criterion.

**Figure 4 ijerph-20-03253-f004:**
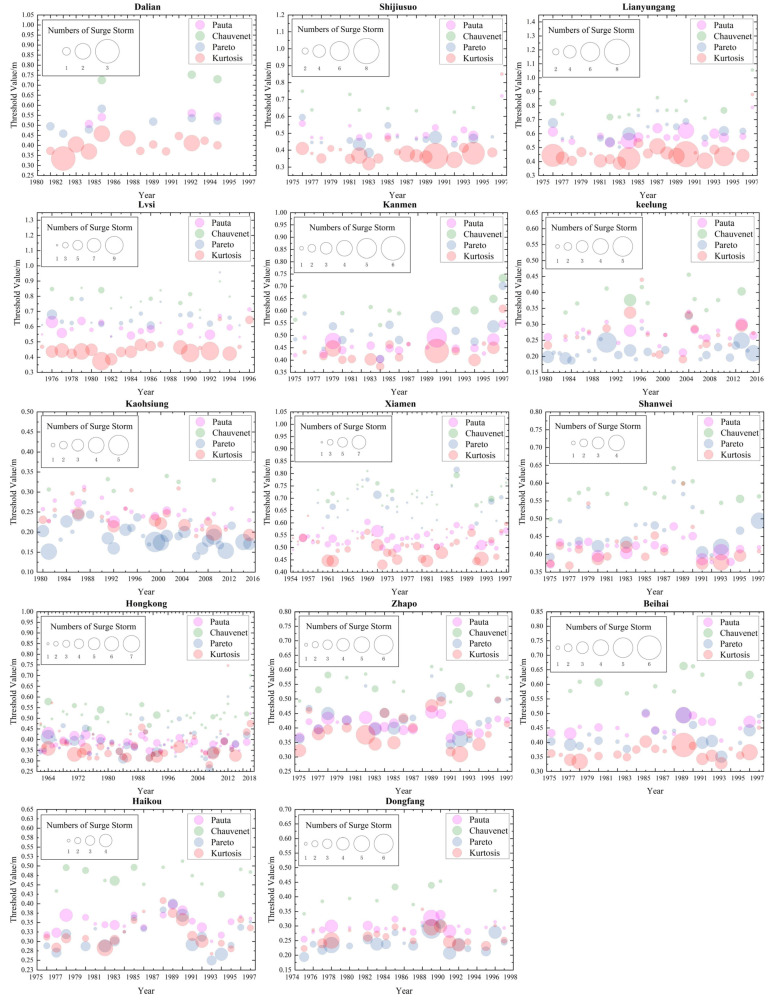
Comparison of the thresholds calculated by the four methods at each station and the numbers of detected storm surges. Note: The abscissa represent the year, whereas the ordinate represents threshold (m). Different colors represent different methods: magenta (pink) for PC, green for CC, blue for PD and red for KC. The size of the circle (◯) corresponds to the number of detected storm-surge events and the y-axis value corresponding to location of circle center is the size of the threshold calculated by a particular method for that year.

**Figure 5 ijerph-20-03253-f005:**
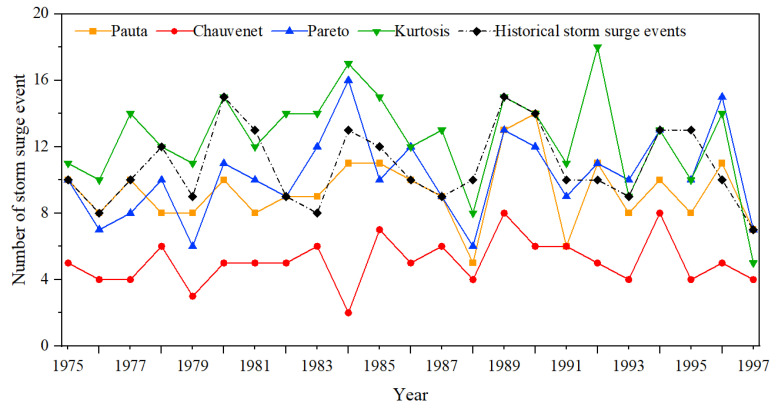
Numbers of annual storm-surge events detected by the four methods from 1975 to 1997.

**Figure 6 ijerph-20-03253-f006:**
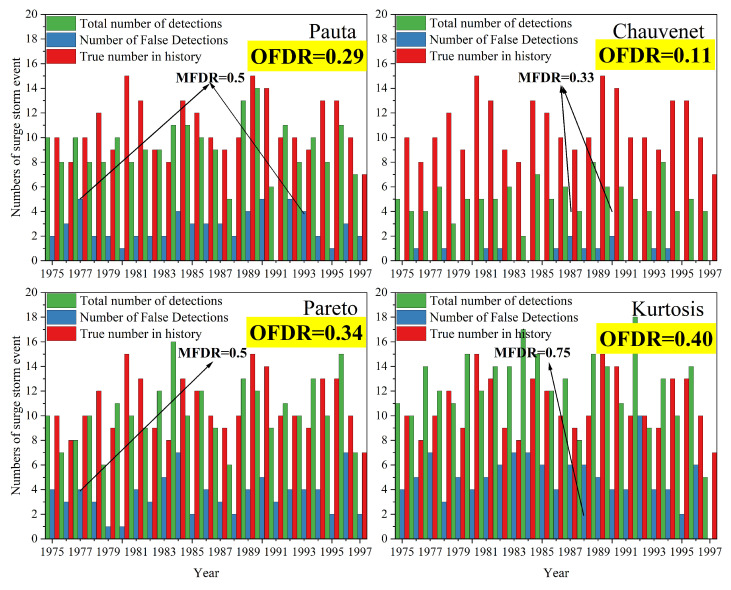
Comparison histograms of the number of storm-surge events detected by the four methods from 1975 to 1997 (green bars, labeled ‘Total number of detections’), the number of error storm-surge events (blue bars, labeled ’Number of False Detections’) and the number of historical storm-surge events (red bars, labeled ’True number in history’). (Note: MFDR is an abbreviation for the maximum false-detection rate. The OFDR is an abbreviation for the overall false detection rate). The end of the arrow represents the year of occurrence of the data at the top of the arrow.

**Table 1 ijerph-20-03253-t001:** Thresholds calculated by the four methods at each tide-gauge station and statistics for detected storm-surge events.

TG_Name	Province	PC		CC		PD		KC	
		Threshold	Storm-Surge Number	Threshold	Storm-Surge Number	Threshold	Storm-Surge Number	Threshold	Storm-Surge Number
Dalian	Liaoning	0.5708 (0.5809)	4 (4)	0.7657 (0.7478)	3 (3)	0.5396 (0.5259)	7 (7)	0.4039 (0.4288)	20 (20)
Shijiusuo	Shandong	0.4924 (0.5032)	20 (25)	0.6605 (0.6781)	6 (8)	0.4724 (0.4774)	25(29)	0.379 (0.4015)	59 (70)
Lianyungang	Jiangsu	0.5698 (0.5789)	37 (44)	0.7644 (0.7805)	12 (16)	0.6332 (0.6289)	25 (30)	0.4491 (0.4654)	67 (85)
Lusi	Jiangsu	0.5787 (0.5817)	40 (58)	0.7763 (0.7719)	17 (22)	0.6702 (0.65)	29 (41)	0.456 (0.4529)	88 (116)
Kanmen	Zhejiang	0.4439 (0.4503)	26 (36)	0.5955 (0.6137)	11 (15)	0.5073 (0.5189)	20 (27)	0.4425 (0.4564)	29 (38)
Keelung	Taiwan	0.2664 (0.2671)	12 (25)	0.3574 (0.3691)	7 (13)	0.2205 (0.214)	27 (51)	0.2881 (0.2717)	11 (24)
Kaohsiung	Taiwan	0.2634 (0.2497)	14 (28)	0.3534 (0.3357)	3 (6)	0.2195 (0.2008)	33 (80)	0.282 (0.245)	16 (40)
Xiamen	Fujian	0.5358 (0.5366)	37 (101)	0.7189 (0.7211)	14 (34)	0.6921 (0.6877)	17 (43)	0.5217 (0.5249)	50 (119)
Shanwei	Guangdong	0.4138 (0.4145)	32 (42)	0.555 (0.5592)	12 (17)	0.4601 (0.4609)	25 (33)	0.4418 (0.417)	29 (39)
Hongkong	Hongkong	0.3809 (0.3802)	29 (104)	0.511 (0.514)	11 (46)	0.3903 (0.3946)	33 (101)	0.3873 (0.39)	39 (122)
Zhapo	Guangdong	0.414 (0.4133)	47 (62)	0.5555 (0.5495)	16 (22)	0.4471 (0.4432)	40 (53)	0.4076 (0.3973)	51 (66)
Beihai	Guangxi	0.4525 (0.4504)	25 (32)	0.607 (0.6082)	12 (14)	0.4234 (0.4227)	30 (39)	0.3674 (0.3681)	39 (51)
Haikou	Hainan	0.3501 (0.3486)	32 (42)	0.4697 (0.468)	16 (20)	0.3208 (0.3202)	43 (52)	0.3302 (0.3278)	37 (47)
Dongfang	Hainan	0.2989 (0.2961)	30 (40)	0.401 (0.3978)	9 (11)	0.2489 (0.2466)	45 (60)	0.2738 (0.2685)	37 (50)

Note: The data corresponding to the column of thresholds outside the brackets are the mean thresholds of each station during the overlapping years (1980–1996) and the data inside the brackets are the mean thresholds of the station in all years. The data corresponding to the column of the number of storm surges outside the brackets are the sum of the storm-surge events detected by each station during the overlapping years (1980–1996) and the data inside the brackets are the sum of the storm-surge events detected by the station in all years.

**Table 2 ijerph-20-03253-t002:** Calculation results of evaluation indicators for the four methods.

Year	PC			CC			PD			KC		
	P	R	F1	P	R	F1	P	R	F1	P	R	F1
1975	0.80	0.80	0.80	1.00	0.50	0.67	0.60	0.60	0.60	0.64	0.70	0.67
1976	0.63	0.63	0.63	0.75	0.38	0.50	0.57	0.50	0.53	0.50	0.63	0.56
1977	0.50	0.50	0.50	1.00	0.40	0.57	0.50	0.40	0.44	0.50	0.70	0.58
1978	0.75	0.50	0.60	0.83	0.42	0.56	0.70	0.58	0.64	0.75	0.75	0.75
1979	0.75	0.67	0.71	1.00	0.33	0.50	0.83	0.56	0.67	0.55	0.67	0.60
1980	0.90	0.60	0.72	1.00	0.33	0.50	0.91	0.67	0.77	0.73	0.73	0.73
1981	0.75	0.46	0.57	0.80	0.31	0.44	0.60	0.46	0.52	0.58	0.54	0.56
1982	0.78	0.78	0.78	0.80	0.44	0.57	0.67	0.67	0.67	0.57	0.89	0.70
1983	0.78	0.88	0.82	1.00	0.75	0.86	0.58	0.88	0.70	0.50	0.88	0.64
1984	0.64	0.54	0.58	1.00	0.15	0.27	0.56	0.69	0.62	0.59	0.77	0.67
1985	0.73	0.67	0.70	1.00	0.58	0.74	0.80	0.67	0.73	0.60	0.75	0.67
1986	0.70	0.70	0.70	0.80	0.40	0.53	0.67	0.80	0.73	0.67	0.80	0.73
1987	0.67	0.67	0.67	0.67	0.44	0.53	0.67	0.67	0.67	0.54	0.78	0.64
1988	0.60	0.30	0.40	0.75	0.30	0.43	0.67	0.40	0.50	0.25	0.20	0.22
1989	0.69	0.60	0.64	0.88	0.47	0.61	0.69	0.60	0.64	0.67	0.67	0.67
1990	0.64	0.64	0.64	0.67	0.29	0.40	0.58	0.50	0.54	0.71	0.71	0.71
1991	1.00	0.60	0.75	1.00	0.60	0.75	0.67	0.60	0.63	0.64	0.70	0.67
1992	0.55	0.60	0.57	1.00	0.50	0.67	0.64	0.70	0.67	0.44	0.80	0.57
1993	0.50	0.44	0.47	0.75	0.33	0.46	0.60	0.67	0.63	0.56	0.56	0.56
1994	0.80	0.62	0.70	0.88	0.54	0.67	0.69	0.69	0.69	0.69	0.69	0.69
1995	0.88	0.54	0.67	1.00	0.31	0.47	0.80	0.62	0.70	0.80	0.62	0.70
1996	0.73	0.80	0.76	1.00	0.50	0.67	0.53	0.80	0.64	0.57	0.80	0.67
1997	0.71	0.71	0.71	1.00	0.57	0.73	0.71	0.71	0.71	1.00	0.71	0.83
Overall	0.71	0.61	0.66	0.89	0.42	0.57	0.66	0.62	0.64	0.60	0.69	0.65

Note: P represents precision, R represents recall and F1 is a harmonic mean value of P and R, as an evaluation indicator that integrates the two indicators.

**Table 3 ijerph-20-03253-t003:** Comparison of typhoon-storm-surge events detected by four methods with typical storm-surge events in the reference [29] from 1975 to 1997.

Storm-Surge Level	Number in the Book	PC		CC		PD		KC	
		DN	DR	DN	DR	DN	DR	DN	DR
Red	15	15	1	13	0.87	12	0.8	15	1
Orange	36	33	0.92	26	0.72	32	0.89	31	0.86
Yellow	27	23	0.85	20	0.74	24	0.89	23	0.85
Blue	6	5	0.83	4	0.67	5	0.83	5	0.83
Overall	84	76	0.90	63	0.75	73	0.87	74	0.88

Note: DN is an abbreviation for detection number; DR is an abbreviation for detection ratio.

## Data Availability

The data are freely available upon request.

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
