# Peer review of "Comparison and Analysis of Detection Methods for Typhoon-Storm Surges Based on Tide-Gauge Data—Taking Coasts of China as Examples"

_ijerph, 2023, doi:10.3390/ijerph20043253_

Round 1

Reviewer 1 Report

This manuscript detected the storm surge events using four detection methods from the hourly residual water level data of 14 tide gauges along the coasts of China. In my opinion, the manuscript is written well with significant work.

This manuscript includes but does not limited to the following questions:

1. Literature review needs to include more support for the use of four common outlier detection methods, such as the Pauta criterion (PC), the Chauvenet criterion (CC), the Pareto distribution (PD) and the kurtosis coefficient (KC) , in order to better support the significance of this study.

2. At times, typos distract the reader's attention. These need to be thoroughly corrected.

3.In 3.2.1, the number of storm surge events should be summarized with different detection methods, e.g., according to the time and location, how a storm surge event is judged? In addition, what is the radius of the location?

4. The sub-figures in Figure 4 are too small to see clearly, it is recommended to adjust the layout. At the same time, the boundary of Figure 5 seems to be not standardized and needs to be readjusted.

Reviewer 2 Report

This manuscript used four common outlier detection methods to detect storm surge events from the hourly residual water level data of 14 tide gauges along the coasts of China, which includes but is not limited to the following questions:

1. It should be noted that this manuscript does not describe much about the application and references of the four common outlier detection methods in the introduction section. I think it is necessary to introduce the pros and cons of current outlier detection methods in the introduction and explain why these four detection methods were selected for this study.

2. In the 3.2.1, it is necessary to statistically summarize the number of storm surge events obtained by different detection methods, so how to judge a storm surge event according to time and location? What is the geographical size of the location? Is the time within a few hours in a row? Please elaborate.

3. The references are relatively outdated, and the relevant research results in the past three years should be added. You should also increase some comparative analysis of countries to attract more readers.

4. It is recommended to label the title of each detection method in 2.2.2 with a numeric number, like 1...2...3...in 2.2.1. At the same time, the sub-figures in Figure 4 are too small to see clearly, it is recommended to adjust the layout.
